

# Ginkgetin attenuates bone loss in OVX mice by inhibiting the NF-κB/IκBα signaling pathway

GeJin Wei[1,*], Xiongbo Liang[2,*], Feng Wu[1], Changzhi Cheng[1], Shasha Huang[1] and Yanping Zeng[1]

[1] Department of Orthopaedics, 923rd Hospital of PLA, Nanning, China
[2] Graduate School, Guilin Medical College, Guilin, China
* These authors contributed equally to this work.

Corresponding author
Yanping Zeng, 627988989@qq.com

## ABSTRACT

**Background:** Osteoporosis is a disease associated with bone resorption, characterized primarily by the excessive activation of osteoclasts. Ginkgetin is a compound purified from natural ginkgo leaves which has various biological properties, including anti-inflammation, antioxidant, and anti-tumor effects. This study investigated the bone-protective effects of ginkgetin in ovariectomized (OVX) mice and explored their potential signaling pathway in inhibiting osteoclastogenesis in a mouse model of osteoporosis.

**Methods:** Biochemical assays were performed to assess the levels of Ca, ALP, and P in the blood. Micro CT scanning was used to evaluate the impact of ginkgetin on bone loss in mice. RT-PCR was employed to detect the expression of osteoclast-related genes (ctsk, c-fos, trap) in their femoral tissue. Hematoxylin and eosin (H&E) staining was utilized to assess the histopathological changes in femoral tissue due to ginkgetin. The TRAP staining was used to evaluate the impact of ginkgetin osteoclast generation *in vivo*. Western blot analysis was conducted to investigate the effect of ginkgetin on the expression of p-NF-κB p65 and IκBα proteins in mice.

**Results:** Our findings indicate that ginkgetin may increase the serum levels of ALP and P, while decreasing the serum level of Ca in OVX mice. H&E staining and micro CT scanning results suggest that ginkgetin can inhibit bone loss in OVX mice. The TRAP staining results showed ginkgetin suppresses the generation of osteoclasts in OVX mice. RT-PCR results demonstrate that ginkgetin downregulate the expression of osteoclast-related genes (ctsk, c-fos, trap) in the femoral tissue of mice, and this effect is dose-dependent. Western blot analysis results reveal that ginkgetin can inhibit the expression of p-NF-κB p65 and IκBα proteins in mice.

**Conclusion:** Ginkgetin can impact osteoclast formation and activation in OVX mice by inhibiting the NF-κB/IκBα signaling pathway, thereby attenuating bone loss in mice.

# INTRODUCTION

Osteoporosis is one of the most common diseases among elderly individuals and postmenopausal women (*LeBoff et al., 2022*). It significantly affects the quality of life of

patients (*Hernlund et al., 2013*; *Compston, McClung & Leslie, 2019*) and often results in pain and carries a high risk of fractures (*Hernlund et al., 2013*). It has been reported that in the osteoporotic population over 50 years of age, the fracture rate is approximately 40–50% in women and 13–22% in men, with associated disability and increased mortality rates (*Tom et al., 2013*). In East Asian countries, birth rates are continuously decreasing, and the aging population is growing, leading to a substantial increase in the number and proportion of patients with osteoporosis. Therefore, the treatment of osteoporosis has become increasingly important and urgent.

Bone homeostasis is maintained by the dynamic balance between bone formation and bone resorption (*Wang et al., 2015*). Imbalance in bone remodeling, especially enhanced osteoclast activity, is a major cause of osteoporosis. Inhibiting the abnormal bone resorption capacity of osteoclasts may be a significant therapeutic strategy for treating osteoporosis (*Novack & Teitelbaum, 2008*). The proliferation, differentiation, and activation of osteoclasts require the induction of Receptor Activator of Nuclear Factor Kappa-B Ligand (RANKL) (*Kim et al., 2020*). The NF-κB signaling pathway is involved in regulating osteoclast proliferation and functional activation (*Qian et al., 2020*). During the process of bone resorption, RANKL induces the activation of NF-κB, degradation of IκB (inhibitor of NF-κB), and phosphorylation of p65 (a subunit of NF-κB), subsequently activating the major transcription factor NFATc1 for osteoclast formation (*Wei et al., 2021*). Activated NFATc1 induces the expression of osteoclast-specific genes, including c-fos, trap, and cathepsin K (ctsk), which are crucial for osteoclast differentiation and functional activation (*Park, Lee & Lee, 2017*). Therefore, targeting the RANKL-mediated NF-κB signaling pathway to inhibit osteoclast formation may represent a potential therapeutic strategy for bone-related diseases.

Approved drugs that may reportedly inhibit bone resorption include hormone replacement therapy (HRT), bisphosphonates, estrogen agonists, and salmon calcitonin (*Arceo-Mendoza & Camacho, 2021*). However, these drugs exhibit related adverse effects. For instance, HRT increases the risk of breast cancer, cardiovascular diseases, and biliary tract diseases; bisphosphonates may lead to osteonecrosis of the jaw (*Black et al., 2019*), limiting their widespread clinical application. Therefore, there is growing interest in using natural herbal remedies with minimal side effects to inhibit bone resorption. Ginkgetin, a compound purified from natural ginkgo leaves, possesses various biological activities, including anti-inflammatory, antioxidative, and anti-tumor effects. Ginkgetin is considered a natural compound with significant potential (*Guo et al., 2020*). *Lim et al. (2006)* found that ginkgetin can inhibit the expression of NF-κB. *Zhou et al. (2011)* demonstrated that ginkgetin can inhibit the NF-κB signaling pathway and regulate the expression of inflammatory cytokines, chemokines, and immune receptors. NF-κB is considered to be the fundamental pathway for RANKL-induced osteoclast formation and maturation. Inspired by these findings, we hypothesize that ginkgetin has inhibitory effects on osteoclast formation and activation. This study aims to investigate the impact of ginkgetin on bone loss in a mouse model of osteoporosis and its potential mechanisms at the molecular level.

## MATERIALS AND METHODS

### Reagents and materials

Ginkgetin was obtained from the Shanghai Plant Standardization Biological Company, and prostaglandin E2 (catalog no: HY-101952) was purchased from MedChemExpress (MCE). The ELISA kits were purchased from Elabscience Biotechnology Co., Ltd., Houston, TX, USA. The following reagents were used: SYBR Green PCR Kit (F-415XL) and Reverse Transcription Kit (K1622; Thermo Fisher Scientific, Waltham, MA, USA). Trizol was acquired from Invitrogen, DEPC-treated water was procured from Sigma, and chloroform/isopropanol/anhydrous ethanol were purchased from the Shanghai National Pharmaceutical Group Chemical Reagent Co., Ltd. Antibodies included P65 (catalog no: 8242T; 1:1,000), p-P65 (catalog no: 3033T; 1:1,000), IKBα (catalog no: 4812S; 1:1,000) obtained from Cell Signaling Technology (CST, Danvers, MA, USA), and GAPDH (catalog no: 60004-1-Ig; 1:5,000) from Proteintech. Horseradish peroxidase-conjugated goat anti-rabbit IgG (catalog no: ZB2301; 1:5,000) and goat anti-mouse IgG (catalog no: ZB2305; 1:5,000) were purchased from Zhongshan Golden Bridge.

### Acute toxicity test

Female C57BL/6J mice (weight: 20–25 g, 16 weeks old) were obtained from Suzhou Sibeifu Biotechnology Co., Ltd., and were housed in separate cages and environment at $20 \pm 2\,°C$ under standard conditions at 55% humidity and a day/night cycle of 12/12 h. Mice were allowed to acclimate to their environment for 2 weeks, simulating a sterile black and white alternating diet. All procedures were conducted in accordance with the regulations outlined in the "Experimental Animal Management Regulations." Four different doses of ginkgetin (20, 10, 5, and 0 mg/ml) were prepared. Twenty-four female C57BL/6J mice were randomly divided into four groups with six mice in each group. Each group was given 250 μL of ginkgetin *via* oral administration. The sample size was estimated based on the law of diminishing returns. No mice were excluded during the experiment. Prior to dosing, the mice were fasted for 12 h but had access to water. Following dosing, the general condition of the mice was observed over 7 days, which included the monitoring of excrement, changes in body weight, secretions, diet, behavior, and fur. The mice were assessed for organ tissue damage and all mice were euthanized by inhaling an excess of 5% isoflurane in this experiment.

### Osteoporosis model

Female C57BL/6J mice were anesthetized with isoflurane and a midline incision was made in the lower abdomen. The abdominal muscles were bluntly separated, the peritoneum was incised, and the abdominal cavity was entered. The bilateral ovaries and surrounding fat were removed from the OVX mice and the incision was sutured. A total of 36 mice were randomly divided into six groups ($n = 6$ per group): (1) sham group. The sham operation group had their ovaries exposed but the surrounding fat tissue was removed before suturing the incision; (2) OVX group; (3) OVX+ ginkgetin (low) group. Low doses of ginkgetin (25 mg/kg) were orally administered to this group; (4) OVX+ ginkgetin (mid) group. Medium doses of ginkgetin (50 mg/kg) were orally administered to this group;

(5) OVX+ ginkgetin (high) group. High dose ginkgetin (100 mg/kg) was orally administered to this group; (6) OVX+PGE2 group. The prostaglandin E2 group received subcutaneous injections of 6 mg/kg prostaglandin E2 once daily. All mice received intraperitoneal injections of penicillin (penicillin (1,600,000 U per bottle) dissolved in 5 mL), 0.2 mL/day, for three consecutive days. After successful establishment of the OVX model, different concentrations of ginkgetin (25, 50, 100 mg/kg) were orally administered to three groups of model mice, twice a week for 8 weeks. Another group of six model mice and six sham-operated mice received an equivalent volume of saline by oral gavage, twice a week for 8 weeks. The prostaglandin E2 group received subcutaneous injections of 6 mg/kg prostaglandin E2 once daily for 8 weeks. Then all mice were euthanized by inhaling excessive isoflurane, and their bones were separated and stored for the following tests. Ethical approval for the use of animals in our scientific research were granted by Ethics Committee of the 923rd Hospital of PLA (NO. 923LL-KY-2024LW-001-01) and Nanning Zhuoqiang Biotechnology (NO. ZQIA-22022-009). All procedures involving animals strictly adhered to ethical standards for laboratory animal research.

## H&E staining
Femoral tissue was immersed in 10% formalin for overnight fixation and was subsequently rinsed with distilled water. The samples were then decalcified in 14% EDTA solution, dehydrated through a graded ethanol series, embedded in paraffin, and sectioned at 4 μm thickness. The sections were placed in a 37 °C incubator for 15 min, deparaffinized in xylene, and gradually rehydrated in a series of ethanol solutions (100%, 95%, 85%, 75%) and distilled water. Staining was performed with hematoxylin for 5 min, followed by treatment with 1% hydrochloric acid and counterstaining with eosin. The samples were dehydrated in ethanol and mounted with neutral resin. Tissue morphology was observed and photographed at 200× magnification under an optical microscope (Olympus Optical Co. Ltd., Tokyo, Japan) for H&E staining of the femoral tissue sections from each group of mice.

## TRAP staining
TRAP staining was performed according to the manufacturer's instructions, using a commercial staining kit. The femoral sections were deparaffinized, followed by TRAP incubation solution preparation, incubation staining, hematoxylin staining, routine dehydration, and neutral gum sealing. The TRAP staining tissue morphology of the femur tissue slices of each group of mice were observed and photos were taken using an optical microscope at 200×. Quantitative analyses were performed using ImageJ 9.0 software.

## Biochemical analysis and enzyme-linked immunosorbent assay (ELISA)
All mice were euthanized in a closed chamber with 5% isoflurane 8 weeks post-operation. Approximately 3 mL of blood was collected *via* retro-orbital sampling. Blood samples were centrifuged at 3,000 rpm, and the serum was extracted and stored at −20 °C. Biochemical analysis of blood serum for Ca, ALP, and P content was performed using a biochemical

analyzer. The expression levels of mouse type I procollagen amino-terminal pro-peptide (PINP), and mouse type I collagen C-terminal peptide (CTX-I) were measured using ELISA. CTXI, and PINP were measured in duplicate using commercially available immunoassay (ELISA) kits which were purchased from Elabscience Biotechnology Co., Ltd., Houston, TX, USA. The detection steps were carried out according to the operating manual. Finally, the absorbance (450 nm) was measured by means of a microplate reader. For calculating the concentration of CTX-I (ng/mL), a standard curve was drawn.

## Micro CT scanning analysis

The bilateral tibias were collected, separated and the excess surface soft tissue was removed. The samples were fixed in 4% paraformaldehyde (PFA). After 24 h, the PFA was removed, and the tibiae were washed twice with PBS. The tibiae were placed in EP tubes and sent to Zhuoqiang Biotechnology Co., Ltd. (Nanning, Guangxi, China) for high-resolution Bruker micro-CT (Skyscan-1176; Skyscan, Kontich, Belgium) scanning. Scans were performed using 8.96 μm voxel size, 50 KV, 450 μA and 0.4 degrees rotation step (180 degrees angular range). A region of interest (0.5 mm below the growth plate on the proximal tibia with a height of 2 mm) was selected for trabecular bone analysis. Hydroxyapatite phantoms of 0.25 and 0.75 g/cm$^3$ were used and scanned according to the scan protocol to perform a bone mineral density (BMD) calibration with respect to the attenuation values. The index including directly measured bone volume fraction (BV/TV), number (Tb.N) and separation (Tb.Sp) were calculated for the trabecular bone. NR Econ software (version 1.6) was used for 3D reconstruction and viewing of images. After 3D reconstruction, bone analysis was conducted using CT software.

## RT-PCR to detect changes in target gene expression

Approximately 100 mg of femoral tissue from each group was placed in a mortar and ground into a powder with liquid nitrogen. 1 mL of Trizol (Invitrogen, Waltham, MA, USA) was added to fully dissolve the tissue, and the mixture was further homogenized. The homogenate was then transferred to RNase-free 1.5 mL EP tubes and incubated for 10 min. Total RNA was extracted and reverse transcribed into cDNA using Primescript RT kits (Thermo Fisher Scientific, Waltham, MA, USA) according to the manufacturer's instructions. The reverse transcription procedure used the following mixture: 1 ml of random primer p(dN)6 (0.2 μg/μl), 5 ml of Rnase-free ddH$_2$O, 4 ml of 5 × reaction buffer, 2 ml of dNTP Mix (10 mM), 1 ml of Rnase inhibitor (20 U/μl), 2 ml of AMV Reverse Transcriptase (10 U/μl) and 5 ml of RNA. PCR amplification was performed under the following conditions: pre-denaturation at 94 °C for 10 min, followed by 40 cycles of denaturation at 94 °C for 20 s, annealing at 55 °C for 20 s, and extension at 72 °C for 20 s. A final extension step at 72 °C for 10 min was performed. The $2^{-\Delta\Delta Ct}$ method was used for calculating the relative transcription level of the target gene. qRT-PCR was performed using Real-time PCR instrument (ABI-7500). PCR primer sequences can be found in Table 1.

**Table 1 PCR primer sequences.**

| Primer | Sequence(5′-3′) | Product |
|---|---|---|
| CTSK(Mouse)-R | TGTGACCGTGATAATGTGAA | 150 bp |
| CTSK(Mouse)-F | GCAGGCGTTGTTCTTATTC | |
| Trap (mouse)-F | TCCCCAGCCCTTACTACCGTTT | 144 bp |
| Trap (mouse)-R | CTCCCAGGTCTCGAGGCATTTT | |
| C-Fos(Mouse)-F | TGAAGACCGTGTCAGGAG | 167 bp |
| C-Fos(Mouse)-R | CGCTTGGAGTGTATCTGTC | |
| NFATc1(Mouse)-F | CCGAGGAAGAACACTACAG | 145 bp |
| NFATc1(Mouse)-R | GGATGATTGGCTGAAGGAA | |
| Gapdh(Mouse)-F | GGTGAAGGTCGGTGTGAACG | 233 bp |
| Gapdh(Mouse)-R | CTCGCTCCTGGAAGATGGTG | |

## Western-blot

Approximately 100 mg of femoral tissue was placed in a mortar, followed by the addition of a suitable amount of liquid nitrogen for grinding. The tissue was then transferred to a 2 mL homogenizer, and 500 μL of pre-cooled RIPA lysis buffer (containing PMSF) was added for homogenization, ensuring thorough tissue disruption. After 30 min, the lysate was moved to 1.5 mL centrifuge tubes and centrifuged at 12,000 rpm for 10 min at 4 °C. The supernatant was collected as the protein extract. The protein concentration in the samples was immediately determined using a BCA protein concentration measurement kit, and the protein concentrations were adjusted to be consistent. An equal volume of 5× SDS loading buffer was added to the samples, followed by denaturation of the proteins by boiling in a water bath at 95 °C. The samples were then centrifuged at 12,000 rpm for 5 min. The resulting protein samples were either directly loaded for analysis or stored at −80 °C. A 12% SDS-polyacrylamide gel was prepared, and protein samples were loaded and subjected to electrophoresis. The proteins were then transferred onto a PVDF membrane. The PVDF membrane was completely covered after blocking with 5% non-fat milk (diluted in TBST) at room temperature on a shaker for 2 h. After blocking, the PVDF membrane was washed three times with TBST, for 5 min each time. The primary antibodies were diluted in blocking solution to the appropriate concentration and then incubated with the membrane at 4 °C overnight (1:1,000). The PVDF membrane was fully covered with the primary antibodies. After incubation, the membrane was washed three times with TBST, each time for 5 min. The secondary antibodies (diluted 1:5,000 in TBST) were applied and incubated at room temperature for 1 h. The membrane was washed three times with TBST, each time for 5 min. The A and B components of the ECL reagent were mixed in a 1:1 ratio by volume and then applied to the membrane. The membrane was incubated at room temperature for 5 min in a darkroom and chemiluminescent images were obtained using a gel imaging system (ChemiScope 5300 Pro; ChemiScope, Darmstadt, Germany). ImageJ image processing software was used to analyze band intensity. Semi-quantitative analysis was performed by calculating the ratio of the intensity

of the target bands to the average intensity of the corresponding reference bands, and the results were presented in bar graphs.

### Statistical analysis

In this study, data for quantitative measurements were expressed as mean ± SD. Data such as cell viability and RT-qPCR results were expressed as percentages. All data were subjected to statistical analysis using SPSS 22 software (IBM Corporation, Armonk, NY, USA). Differences between groups were analyzed using one-way ANOVA followed by Tukey's test. Student's t-test was used to compare differences between two groups, and chi-squared test was used for categorical data. A significance level of $P < 0.05$ was considered to indicate statistical significance.

## RESULTS

### Ginkgetin showed no apparent toxic effects on mice

After the administration of ginkgetin, the animals reduced their activity and became calm. However, within 1 h, the mice's eating and behavior returned to normal. No significant signs of toxicity were observed and the mice were monitored for an additional 3 days. There were no abnormal secretions from the mouth, nose, or eyes in the mice in any of the intervention groups. Moreover, no significant pathological changes or deaths were observed in any of the groups ($P > 0.05$).

### Effects of ginkgetin on serum ALP, P, and Ca levels in OVX mice

Biochemical analysis showed that compared to the sham group, the levels of ALP and P in the blood of ovariectomized mice were significantly increased, while Ca levels were significantly decreased, indicating the successful modeling of osteoporosis in OVX mice (Fig. 1). When compared to the model group, the ginkgetin groups (at doses of 200, 100, and 50 mg/kg) showed no statistically significant differences in their ability to affect the serum Ca, P, and ALP levels ($P > 0.05$).

### Ginkgetin alleviated bone loss in OVX C57BL/6J mice

Micro CT scanning results indicated that the tibial trabeculae appeared plate-like, dense, and uniform in the sham group. In contrast, the tibial trabeculae in OVX mice were sparse. Compared to the OVX mice, the mice treated with ginkgetin at low, medium, and high doses (50, 100, and 200 mg/kg, respectively) showed increased trabecular numbers, trabecular connectivity density, and varying degrees of increased bone density. The bone microarchitecture parameters such as BMD, BV/TV, and Tb.N were significantly improved and Tb.Sp was significantly decreased after ginkgetin administration in OVX mice in a dose-dependent manner (Fig. 2). The quantification of ginkgetin alleviated bone resorption, which led to the protective effect of OVX-associated bone loss.

### Effects of ginkgetin on the morphology of OVX mice bone tissue

H&E staining showed that the tibial trabeculae in the OVX group appeared sparse, structurally incomplete, significantly thinner, disorganized, and contained numerous empty lacunae compared to the sham group. The trabecular connectivity was lost, and the

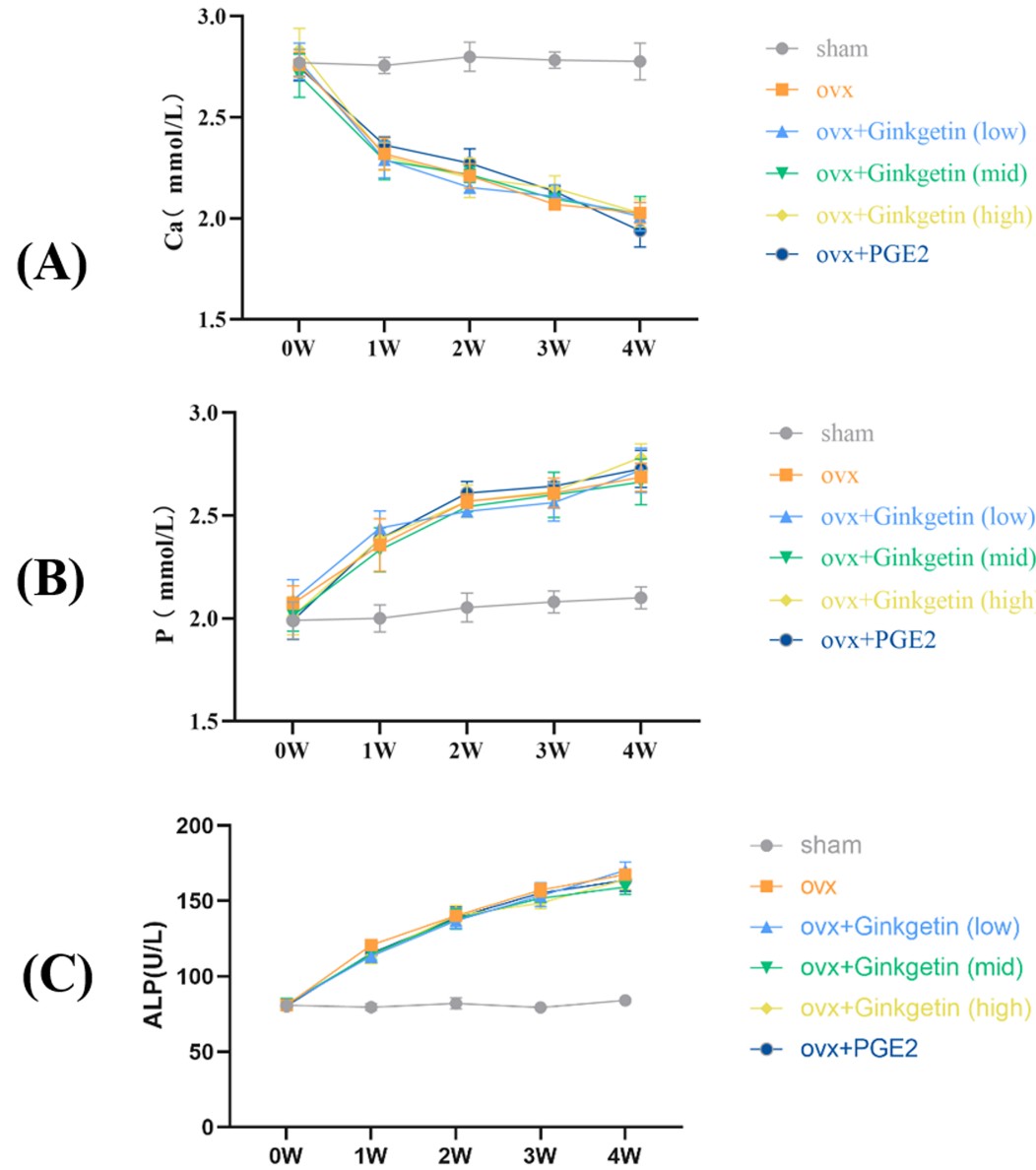

**Figure 1 Effects of ginkgetin on serum ALP, P, and Ca levels measured by the biochemical instrument at 1-, 2-, 3- and 4-week respectively in OVX mice.** (A) The effects of different concentrations of ginkgetin on serum Ca were measured and the levels of serum Ca were significantly decreased. (B) The effects of different concentrations of ginkgetin on serum P were measured and the levels of serum p were significantly increased. (C) The effects of different concentrations of ginkgetin on serum ALP were measured and the levels of serum ALP were significantly increased.

trabecular spaces were widened. When compared to the OVX group, the low-dose ginkgetin group showed some improvement in the tibial trabeculae morphology, while the medium and high-dose groups demonstrated significantly improved tibial trabeculae, with more intact structures and reduced empty lacunae. The high-dose group exhibited the most substantial improvement (Fig. 3).

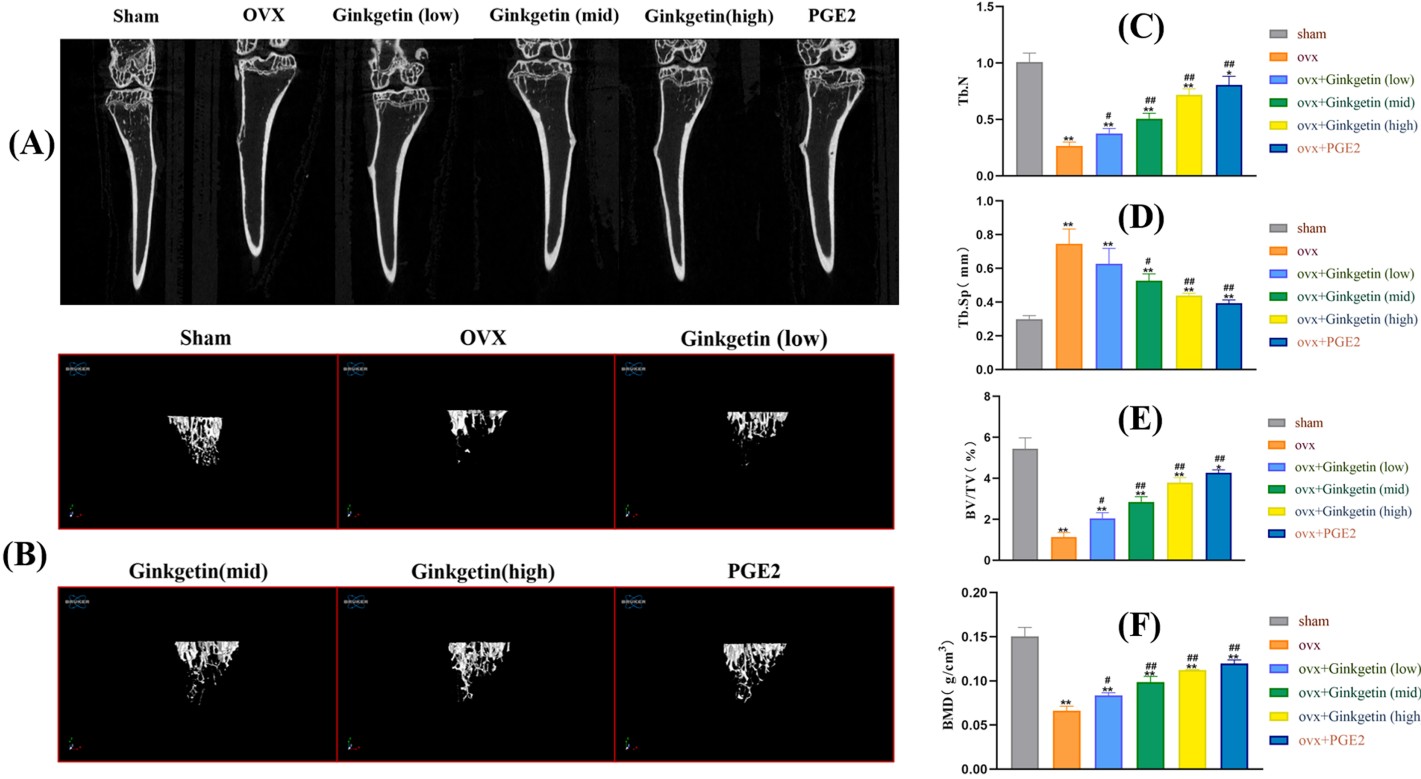

**Figure 2  Ginkgetin has a protective effect on bone loss in OVX C57BL/6J mice.** (A) Representative 2D-μCT images of tibias in different groups of mice. (B) Representative 3D-μCT images of tibias in different groups of mice. (C) Comparative analysis of trabecular bone number (Tb.N) that were significantly improved after ginkgetin administration. (D) Comparative analysis of trabecular bone separation (Tb.Sp, mm) that were significantly decreased after ginkgetin administration. (E) Comparative analysis of bone volume fraction (BV/TV, %) that were significantly improved after ginkgetin administration. (F) Comparative analysis of bone mineral density (BMD, in g/mm$^3$) that were significantly improved after ginkgetin administration. $n = 6$. All bar charts are presented as the mean ± SD. $^*P < 0.05$, $^{**}P < 0.01$ compared with the sham group. $^{\#}P < 0.05$, $^{\#\#}P < 0.01$ compared with the OVX group.                               

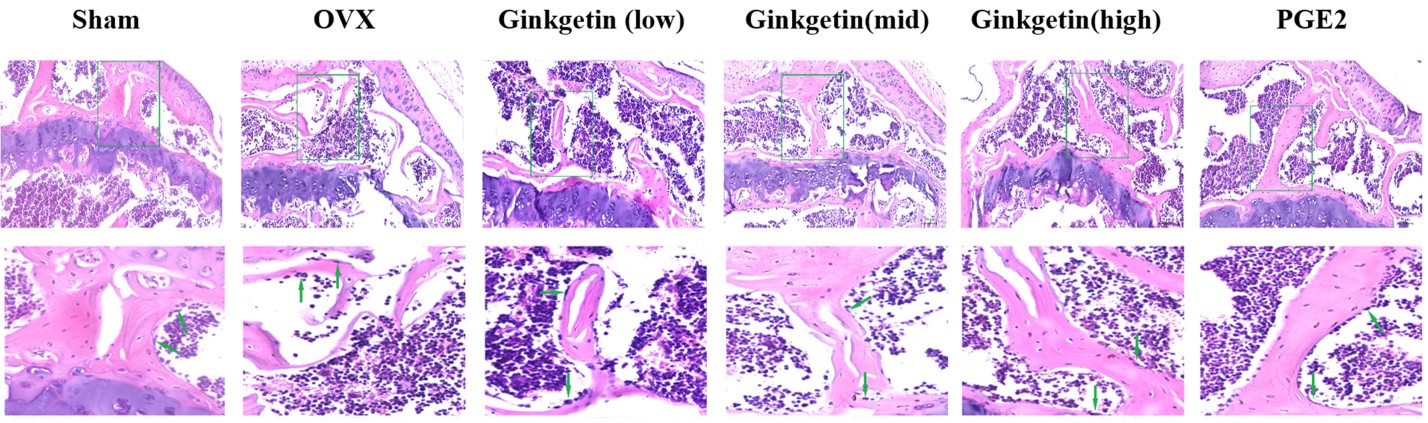

**Figure 3  Effects of ginkgetin on the morphology of OVX mice bone tissue.** Representative H&E staining femoral sections of mice in different groups. H&E staining indicated that the thickness of the tibial trabecula surface was more maintained in the ginkgetin-treated groups than in the OVX group in a dose-dependent manner (scale bar = 100 μm). The green arrows refer to the absorption bay.
                                                                           

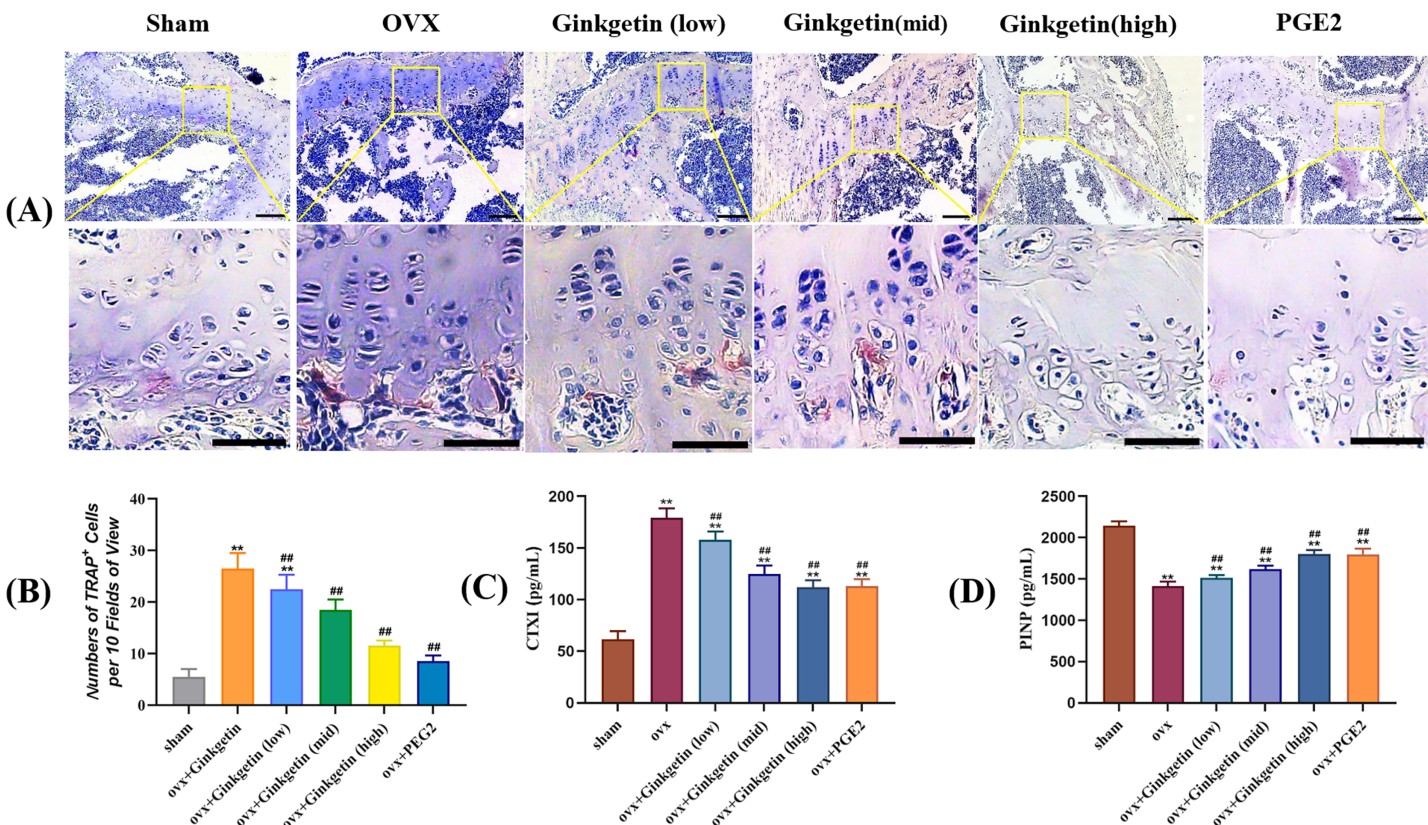

**Figure 4 (A–D) Effects of ginkgetin on the generation of osteoclasts and the expression of bone turnover biomarkers such as CTXI and PINP.** (A) Representative TRAP staining images of osteoclasts in femoral sections treated without or with different concentration ginkgetin. TRAP-positive multinucleated cells (nuclei >3) were regarded as osteoclasts. The original images scale bar = 100 μm, the enlarged images scale bar = 50 μm. (B) Quantitative analysis of TRAP-positive Cells. TRAP staining showing the inhibitory effect of ginkgetin on osteoclastogenesis in a dose-dependent manner. (C) Quantitative analysis of serum CTXI expression. The expression of CTXI was significantly reduced, after ginkgetin intervention with different concentrations. (D) Quantitative analysis of serum PINP expression. The expression of PINP was significantly increased after ginkgetin intervention with different concentrations. **$P < 0.01$ compared with the sham group. ##$P < 0.01$ compared with the OVX group.

## Ginkgetin suppresses the generation of osteoclasts in OVX mice

The ELISA results showed that compared with the sham group, the serum CTXI in the OVX group was significantly increased, while PINP was significantly reduced. After ginkgetin intervention with different concentrations, the expression of CTXI was significantly reduced, while the expression of PINP was significantly increased, in a dose-dependent manner (Figs. 4C and 4D). The TRAP staining showed that the model group had a significant increase in purple-red precipitates in the cytoplasm when compared to the sham group. After the administration of different doses of ginkgetin, the increase was effectively reversed, and the reversal was dose-dependent (Figs. 4A and 4B).

## Ginkgetin suppresses the expression of osteoclast-related genes in OVX mice

We studied whether ginkgetin reduced the mRNA expression of mature osteoclast-related genes to investigate its inhibitory effect on osteoclast formation. Ginkgetin (at doses of 50,

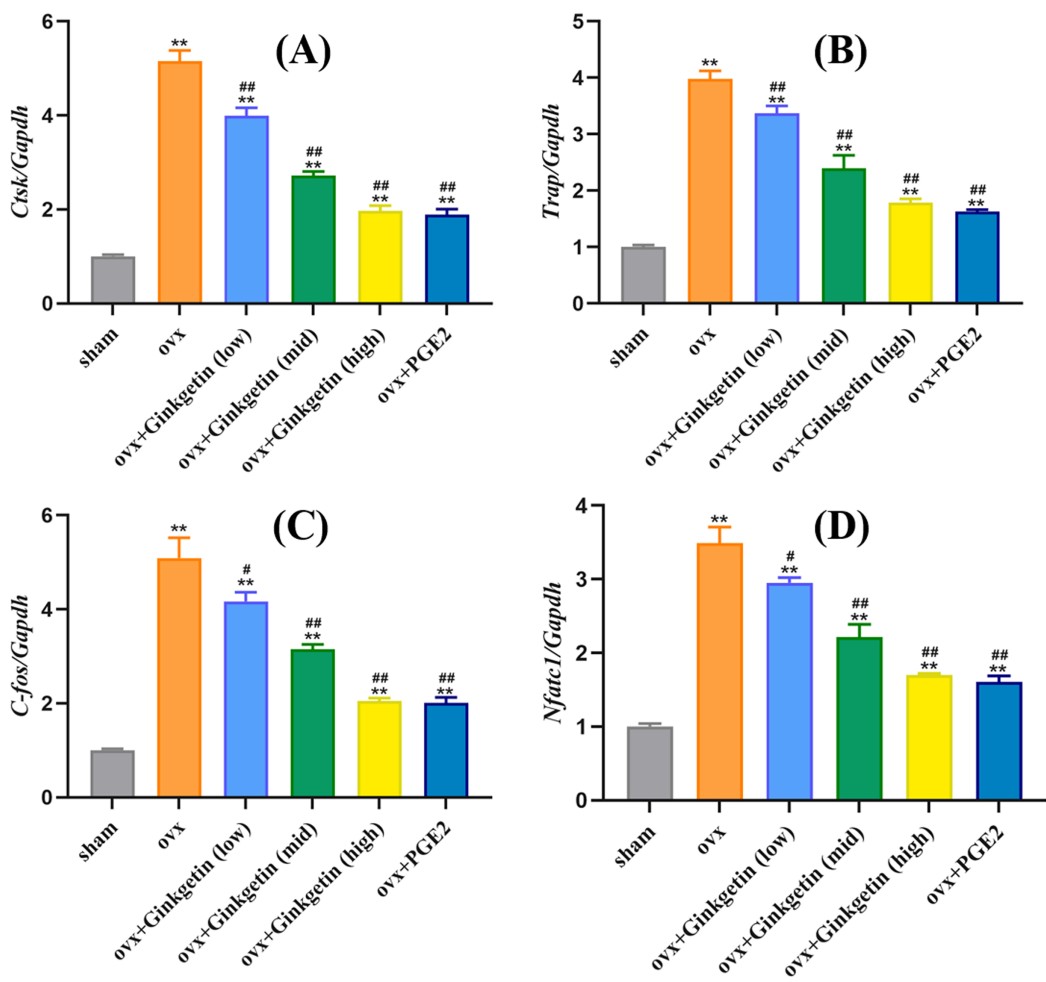

**Figure 5 Ginkgetin suppresses the expression of osteoclast-related genes in OVX mice.** (A) Quantitative analysis of mRNA expression of CTSK. (B) Quantitative analysis of mRNA expression of TRAP. (C) Quantitative analysis of mRNA expression of c-fos. (D) Quantitative analysis of mRNA expression of NFATc1. $^{**}P < 0.01$ compared with the sham group. $^{#}P < 0.05$, $^{##}P < 0.01$ compared with the OVX group.

100, and 200 mg/kg) downregulated the expression of Nfatc1 in a dose-dependent manner (Fig. 5). Additionally, ginkgetin downregulated the expression of osteoclast-related genes such as ctsk, trap, and c-fos (Fig. 3B). These findings suggest that ginkgetin inhibits the expression of osteoclast-specific genes in mice, thereby inhibiting osteoclast formation in a dose-dependent manner.

## Effects of ginkgetin on p-P65 and IKBα

Western blot experiments revealed that compared to the control group (sham), the expression of p-P65 protein was significantly increased in the osteoporosis model, while IKBα expression was significantly decreased. After intervention with different doses of ginkgetin, p-P65 protein expression was significantly reduced, and IKBα expression was significantly increased in a dose-dependent manner (Fig. 6).

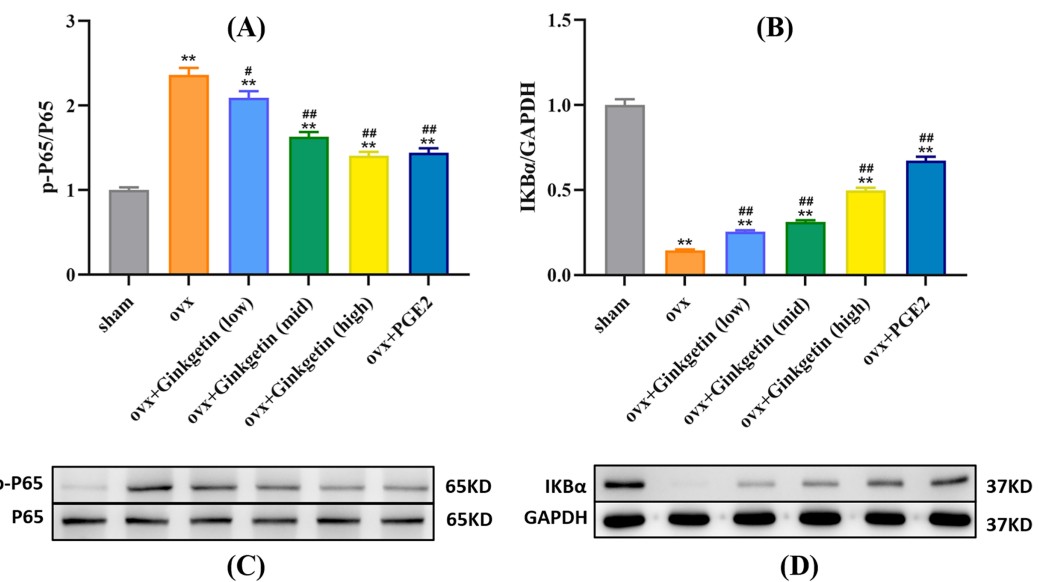

**Figure 6 Different doses of ginkgetin (low, 25 mg/kg; mid, 50 mg/kg; high, 100 mg/kg) suppress the expression of p-P65 and IκBα proteins in C57BL/6 mice.** (A) Histogram showing that ginkgetin inhibits the expression of p-P65 proteins. (B) Histogram showing that ginkgetin inhibits the expression of IκBα proteins. (C) Typical Western blot images of ginkgetin inhibiting the expression of p-P65 proteins. (D) Typical Western blot images of ginkgetin inhibiting the expression of IκBα proteins. $^{**}P < 0.01$ compared with the sham group. $n = 5$, $^{\#}P < 0.05$, $^{\#\#}P < 0.01$ compared with the OVX group.

# DISCUSSION

Approximately one-third of females and one-fifth of males, aged 50 or older, may experience osteoporotic fractures during their lifetime. These occurrences substantially diminish the quality of life for patients, leading to disability and mortality, while imposing a considerable economic burden on families and society (*Sozen, Ozisik & Calik Basaran, 2017*). The onset of osteoporosis is intricately linked to the disrupted dynamic equilibrium between bone formation and resorption. Some studies propose that overactive osteoclasts are responsible for undermining bone microarchitecture, consequently fostering the development of bone loss-related conditions, such as osteoporosis, rheumatoid arthritis, and cancer bone metastasis (*Da, Tao & Zhu, 2021*). Hence, the inhibition of osteoclast activity has become a principal therapeutic strategy for the restoration of bone homeostasis. Recently, first-line anti-osteoporotic medications (*e.g.*, bisphosphonates, parathyroid hormone analogs, selective estrogen receptor modulators, and calcitonin) have demonstrated their efficacy in reducing the risk of osteoporotic fractures. Nevertheless, these pharmaceutical agents exhibit several adverse effects, including an increased risk of atypical femur fractures and jaw necrosis (*Hanley et al., 2017*), gastrointestinal system reactions, and impaired fracture healing post-osteoporotic fracture (*Shibahara, 2019*). Consequently, the development of new, minimally adverse, effective anti-osteoclast drugs is of paramount importance for osteoporosis treatment.

Several natural plant-derived formulations contain an abundance of bioactive chemical compounds that could potentially aid in the treatment of osteoporosis (*Słupski, Jawień &*

*Nowak, 2021*). For centuries, *Ginkgo biloba* leaf extract has been widely used to treat various ailments in Chinese and Western nations, primarily due to its antioxidant properties (*Brinkley et al., 2010*; *Pietri et al., 1997*). Flavonoids represent one of the most crucial active constituents in *Ginkgo biloba* leaf extract and exhibit free radical scavenging, antioxidant, and anti-inflammatory activities (*Vining Smith & Luo, 2003*).

We established an ovariectomy-induced estrogen-deficient mouse model to ascertain whether ginkgetin may be an effective *in vivo* therapy for osteoporosis. By monitoring changes in serum levels of Ca, P, and ALP, we observed an elevated bone turnover rate in accordance with the bone metabolism characteristics of postmenopausal osteoporosis, thereby indicating the successful modeling of ovariectomy-induced estrogen deficiency in mice. The International Osteoporosis Foundation recommends CTXI and PINP as reference biomarkers for bone turnover to assess future fracture risk and evaluate clinical treatment outcomes. PINP is a sensitive biomarker reflecting the dynamic changes in bone formation that reflect the synthesis of new collagen, while CTXI is a sensitive biomarker reflecting the rate of collagen degradation in bone resorption (*Nelson et al., 2020*). Our research showed that compared with the control group, the serum CTXI content in OVX was significantly increased, while the PINP content was significantly reduced. After using different doses of *Ginkgo biloba* flavonoids, the expression of CTXI was significantly reduced, while the expression of PINP was significantly increased.

Bone density serves as a pivotal indicator of bone strength. A decline in BMD levels constitutes a primary cause for the development of osteoporotic diseases and a risk factor for fractures. In our study, we employed micro-CT to reconstruct the three-dimensional structure of tibial trabecular bone tissue in ovariectomy-induced estrogen-deficient mice. In contrast to medical CT, micro-CT can reach micron-level spatial resolution. The regions of interest underwent 2D and 3D reconstruction for quantitative analysis of the bone tissue metrology and spatial ratios. We extracted bone parameters to predict bone strength, aiming to investigate the impact of different concentrations of ginkgetin on *in vivo* bone loss in the OVX model. Our scanning results reveal that, compared to OVX group, tibias of mice treated with ginkgetin had a significant increase in BV/TV, Tb.N, and BMD and a significant decrease in Tb.Sp. Histological examination (H&E staining) demonstrated that the trabecular bone architecture in the distal femur metaphysis of the model group exhibited increased porosity, reduced trabecular number, disorganized arrangement, and poor connectivity following intervention with different concentrations of ginkgetin, compared to the sham surgery group. Effective and dose-dependent reversal of these changes was observed following intervention with different concentrations of ginkgetin. These results indicate a positive role of ginkgetin in preserving bone density in OVX mice.

To further investigate the molecular mechanisms of ginkgetin in combating osteoporosis, we explored the effects of different concentrations of ginkgetin on osteoclasts and signaling pathways in mice. NFATc1 is an indispensable regulatory factor in osteoclast formation and differentiation. Knocking out NFATc1 prevents osteoclast differentiation when stimulated with RANKL, signifying the vital role of NFATc1 in osteoclast formation and function (*Zhao et al., 2021*). Other studies also suggest that inhibiting NFATc1

expression can suppress osteoclast formation (*Dinesh et al., 2020*). These lines of evidence emphasize the necessity of NFATc1 in osteoclast formation and maturation. The components of c-Fos can trigger the activation of the NF-κB pathway, leading to the generation of activator protein-1 (AP-1), thereby promoting NFATc1 expression (*Ono & Nakashima, 2018*). Our findings indicate that ginkgetin can inhibit c-Fos, NFATc1, and other osteoclast-specific genes, such as TRAP and CTSK.

The NF-κB signaling pathway is a fundamental signaling cascade in chronic inflammation, immune responses, and cell apoptosis. It is also a major downstream signaling pathway induced by RANKL for osteoclast differentiation and maturation (*Zhu et al., 2020*). Under normal circumstances, NF-κB proteins remain in an inactive state in the cytoplasm (the p50/p65 heterodimer primarily associates with IκBα, remaining inactive) (*Alharbi et al., 2021*). Typically, nuclear translocation of inactive NF-κB cannot occur until its inhibitor, IκBα, is degraded. Upon stimulation with RANKL, IκBα undergoes phosphorylation, allowing NF-κB p65 to translocate into the nucleus and bind to DNA, thus promoting osteoclast formation and differentiation (*Mockenhaupt, Gonsiewski & Kordula, 2021*). Interestingly, our western blot experiments revealed that IκBα expression was significantly increased in a dose-dependent manner after intervention with different doses of ginkgetin in OVX mice.

Research by *Pan et al. (2019)* suggests that ginkgetin can mitigate autophagy and cell apoptosis induced by cerebral ischemia/reperfusion by inhibiting the NF-κB/p53 signaling pathway. *Cheng, Li & Kong*'s *(2019)* study reveals that ginkgetin exerts its anti-proliferative effects on HeLa cells through a mechanism involving the p38/NF-κB pathway. Our protein imprinting assay results demonstrate that ginkgetin inhibit the phosphorylation and degradation of IκBα within OVX mice and can also suppress p65 phosphorylation. Interestingly, NF-κB is an inflammatory signaling transcription factor that can regulate the inflammatory response, promote the differentiation of osteoclasts and bone remodeling process in bone formation and bone resorption cells. Studies have shown that blocking NF-κB signal transduction can limit inflammation and prevent bone loss (*Lin et al., 2017*; *Tian et al., 2022*). Some limitations are remained in our research. Our study indicates that ginkgetin may play a pivotal role in the regulation of the NF-κB signaling pathway. However, the precise genes targeted by ginkgetin within the NF-κB signaling pathway are not clear, nor what other signaling pathways may be affected. Another limitation is that the voxel size of 8.96 microns used in micro-CT assessment may not provide too much details for mice bone assessment. Therefore, more studies are needed to explore the molecular mechanism of ginkgetin inhibiting bone loss.

## CONCLUSIONS

In summary, our research substantiates that ginkgetin can inhibit bone loss in OVX mice, possibly by suppressing osteoclast formation and maturation through the inhibition of the NF-κB/IκBα signaling pathway. This provides a novel potential therapeutic approach for treating osteoporosis.

### Funding

This work was supported by the Natural Science Foundation of Guangxi Zhuang Autonomous Region [grant number 2018JJA140721]. The funders had no role in study design, data collection and analysis, decision to publish, or preparation of the manuscript.

### Grant Disclosures

The following grant information was disclosed by the authors:
Natural Science Foundation of Guangxi Zhuang Autonomous Region: 2018JJA140721.

### Competing Interests

The authors declare that they have no competing interests.

### Author Contributions

- GeJin Wei conceived and designed the experiments, authored or reviewed drafts of the article, and approved the final draft.
- Xiongbo Liang performed the experiments, analyzed the data, prepared figures and/or tables, and approved the final draft.
- Feng Wu performed the experiments, prepared figures and/or tables, and approved the final draft.
- Changzhi Cheng analyzed the data, prepared figures and/or tables, and approved the final draft.
- Shasha Huang performed the experiments, analyzed the data, prepared figures and/or tables, and approved the final draft.
- Yanping Zeng conceived and designed the experiments, authored or reviewed drafts of the article, and approved the final draft.

### Animal Ethics

The following information was supplied relating to ethical approvals (*i.e.*, approving body and any reference numbers):

This study was granted by Ethics Committee of the 923rd Hospital of PLA (NO. 923LL-KY-2024LW-001-01) and Nanning Zhuoqiang Biotechnology (NO. ZQIA-22022-009).

### Data Availability

The raw measurements are available in the Supplemental File.

### Supplemental Information

Supplemental information for this article can be found online at http://dx.doi.org/10.7717/peerj.17722#supplemental-information.

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
