# Peer review of "Ginkgetin attenuates bone loss in OVX mice by inhibiting the NF-κB/IκBα signaling pathway"

_PeerJ, doi:10.7717/peerj.17722_

## Round 0.1 · original submission · Major Revisions

· Academic Editor

Major Revisions

Please carefully review the feedback provided by the reviewers and make all the necessary changes to the document. All three reviewers have expressed major concerns which should be addressed Once you have made the revisions, please submit the updated version for further review.

Reviewer 1 ·

Basic reporting

The manuscript is written in clear, unambiguous, and professional English, making it accessible to a wide scientific audience. The authors have successfully communicated complex scientific concepts in a comprehensible manner.The study is well-grounded in current scientific literature, demonstrating a thorough understanding of the field. The introduction provides an excellent context for the study, and references are up-to-date and relevant to the research question. The article adheres to a high standard of academic presentation. It is logically structured with an introduction, methods, results, discussion, and conclusion. The figures and tables are of high quality, effectively illustrating key findings. The inclusion of raw data is commendable, enhancing the transparency and reproducibility of the research.The study is self-contained, with a clear hypothesis stated at the outset. The results are relevant and directly address the research question.

Experimental design

The research topic aligns well with the journal's focus and contributes original findings to the field. However, while the research question is clear and relevant, the manuscript could have more explicitly stated how this research fills a specific knowledge gap in the field of bone loss and pharmacology.The investigation appears rigorous, with appropriate technical standards applied. The ethical considerations are addressed, but a more detailed discussion on this aspect would enhance the manuscript's quality.The methods are described with a reasonable level of detail, providing a foundation for replication. However, some sections lack the depth required for thorough reproducibility. Additional information on certain experimental procedures or statistical analyses might be beneficial for readers aiming to replicate the study.The manuscript's scope is appropriate, and the research holds potential significance in its field. Nonetheless, the impact of the findings could be articulated more compellingly, particularly in terms of practical applications or implications for future research.

Validity of the findings

The manuscript falls short in evaluating its own impact and novelty within the area of bone loss research. This lack of assessment hinders a comprehensive appreciation of the study's contribution to the field, particularly in terms of offering new insights or challenging existing paradigms.A strong aspect of the manuscript is the provision of robust, statistically sound, and well-controlled data. The authors commendably encourage replication of their research, providing clear reasons for its importance and potential benefits to the scientific literature. This approach not only supports the validity of their findings but also fosters further exploration in the field.The conclusions of the study are well-formulated and closely tied to the original research question. They are cautiously framed within the bounds of the data presented. However, the manuscript could be enhanced by extending the discussion to explore broader implications and potential applications of the findings, thereby providing a more comprehensive understanding of their significance.

Annotated reviews are not available for download in order to protect the identity of reviewers who chose to remain anonymous.

Reviewer 2 ·

Basic reporting

1. The RGB color scheme used for the histogram is not color-blind friendly. Utilize alternate color schemes.
2. There are several grammatical errors (e.g., in the abstract, line 33 should be Ginkgetin "suppresses" the generation").
3. Correct typos in the article (e.g., Line 259 - "Gdecreased").

I recommend a thorough revision using online tools or assistance from a fluent English speaker.

Experimental design

1. In Figure 1, the explanation and graph do not reveal a significant difference in serum Ca, P, and ALP between Gingketin-treated and untreated groups. Despite this, in the discussion (line 290), the authors mention that high doses decreased serum levels of Ca and ALP. This contradicts the presented data. Clarify what the authors are referring to and provide evidence for reduced bone turnover. If it's an error, rewrite to align with the data.

2. For Figure 3, clearly label the H&E staining images to indicate the location of osteoclast cells and the resorption bay outline. Explain the differences in the purple hue in the Gingketin low-dose and PGE2 images which look drastically different from the other images. Add a detailed explanation to the figure legend, specifying more than just the maintenance of the surface.

3. In Figure 4, to support the statement that Gingketin suppresses osteoclast generation in OVX mice, provide information on changes in expression in bone microenvironment biomarkers such as alkaline phosphatases (BSAP), Osteocalcin, PINP, NTX.

4. In Figure 6, ensure the subheading matches the figures (e.g., Figure 6A states "Typical western blot image" but represents a histogram). Label the western blot clearly with respective doses, as the image lanes are not perfectly aligned with the histogram. Correct the awkward cropping of the 37KD. Explain the dose-dependent increase in IKB protein accumulation in the discussion.

5. Investigate the effects of Ginketin on pro-inflammatory cytokines, considering the association between osteoporosis and inflammation.

Validity of the findings

1. In Figure 4, make it clear that the above panel is an enlarged image of the below panel. Specify this in the legend or use appropriate methods to highlight the enlarged images. Include a quantification of multinucleated cells in different fields to validate the findings. A single representative picture is insufficient.

Reviewer 3 ·

Basic reporting

Narrative Assessment for Authors:

The manuscript titled "Ginkgetin attenuate bone loss in OVX mice by inhibiting the NF-κB/IκBα signaling pathway" investigates whether Ginkgetin has inhibitory effects on osteoclast formation and activation in an animal model. The authors aim to "investigate the impact of Ginkgetin on bone loss in a mouse model of osteoporosis and its potential mechanisms at the molecular level". While the authors put a lot of effort into this study and used many different experimental techniques to analyze the data, the data analysis and data presentation must be improved since this could potentially weaken the conclusions stated in the manuscript. Also, it is not stated that this study aimed to assess the effects of different Ginkgetin concentrations. Extensive English language editing is required. The literature used in the manuscript could be characterized as sufficient to provide the context of the research. The structure of the manuscript is generally acceptable, with a bit of advice to restructure the methodology section (it should be clearly stated which groups were used, the number of animals in each group, and the description of these groups one by one). In my opinion, it would be great if the graphical representation of the study design (with adequate flow chart representation) could be added to the manuscript to ensure a better understanding for the readers.

Experimental design

The experimental design needs further elaboration and better phrasing to be easily comprehensible for the readers. Authors should clearly state which groups were used in the study and add the description of these groups one by one. Also, it would be desirable to be consistent in using the same group names in the text as in the figures. For example, it is not clear from the group description given in the methodology section - osteoporosis model subsection, which group had ovariectomy. I agree that the number of animals in each group should be stated, but also, the total number of animals included in the study should be stated to avoid any misunderstanding by the readers. In my opinion, it would be great if the graphical representation of the study design (with adequate flow chart representation) could be added to the manuscript.
Also, a major concern for the current version of the manuscript is the description of micro-CT analysis. It is advisable to follow current guidelines for micro-CT assessment and reporting in animal models (a good starting point would be in the work by Bouxsein ML et al. published in J Bone Miner Res doi: 10.1002/jbmr.141). Namely, at least scanning protocol parameters including spatial resolution of the scan, should be stated. In the current version of the manuscript, it is only stated that the region of interest was 0.9mm, but that does not mean much because micro-CT analyses 3D bone volume. Also, it is unclear what is meant by the 0.9mm, given that the resolution of the scan determines the accuracy/applicability/relevance of the generated data. Further, it is not clear in any way how the regions of interest were selected for this study. Was it manual, semiautomatic, or automatic protocol? Also, for measuring BMD using micro-CT, it should be stated which (if any) calibrating phantoms were used. From the provided description one could conclude that both tibias and femora were analyzed in each animal, so please rephrase the text to be accurate. If the latter is the case, please provide an explanation in the manuscript. Also, micro-CT can generate much more data than given in the row data table. Thus, it should be explained why only these micro-CT parameters were chosen and why only trabecular bone was analyzed. Also, it is unclear why the authors did not include these data in the figure and provide relevant discussion regarding their data on internal bone micro-architectural parameters. In the row data table, mean and SD values for micro-CT parameters are not presented in a meaningful manner.
Regarding the H&E-stained bone sections, it is unclear which parameters were analyzed. In the results section, only a qualitative description was generated without adequate inclusion of hard data. Also, the assessment of osteocyte lacunar network and bone adiposity parameters could strengthen this article. For more data, please check :
1) Dempster DW et al. J Bone Miner Res. doi: 10.1002/jbmr.1805.
2) Tratwal J et al. Front Endocrinol. doi: 10.3389/fendo.2020.00065.

It is stated that TRAP was done in vivo (line 28), but it is not clear how that was done in the stated study design.

Statistical analysis:
It is unclear what is meant by the sentence: "In this study, each experiment was repeated three times" (line 204). Since this does not seem a reasonable choice for some analyses used in the study, it is required to better explain the study design and/or to use better word choice. Also, the whole statistical analysis section must be better written to accurately illustrate the statistical approach. To the best of my knowledge, ANOVA is the test of choice in comparison of multiple groups (more than two), and post hoc tests are applied to investigate the difference between these groups, so these are not mutually exclusive tests, as stated in the manuscript. Also, only if ANOVA reveals statistical significance post hoc tests (such as the Tukey post hoc test) should be taken into account. Did the authors check for data distribution normality and data variance homogeneity that is required for ANOVA? Also, it is unclear for which part of the study ( which parameter/parameters) the authors used the student t-test and which type of student t-test was used. Individual data points should be added to the figures as well.

Validity of the findings

Data presented in the current version of the manuscript required major editing and better representation to be properly assessed for its validity. Study limitations should be clearly stated in the manuscript to provide additional support for the validity of the data in current literature.

Additional comments

The quality of the English language used in the manuscript must be improved. Some examples where the language should be improved include a title (line 2), lines 21-22, lines 27-29, line 71, etc. Also, extensive rephrasing is advisable in the manuscript, especially in the abstract, to avoid misleading statements (such as lines 17-18, line 20, line 44, lines 71-73, etc.). Past tense should be used when reporting the experiments. Patients-first language should also be used in the manuscript ("patients with osteoporosis" rather than "osteoporotic patients"). The discussion section should also include a paragraph clearly stating the limitations of the present study.

---

## Round 0.2 · Minor Revisions

· Academic Editor

Minor Revisions

Please review and submit a revised version based on the comments from the reviewers.

Reviewer 1 ·

Basic reporting

No comments.

Experimental design

No comments.

Validity of the findings

No comments.

Additional comments

No comments.

Reviewer 3 ·

Basic reporting

Dear Authors,

Thank you for your diligent efforts in significantly improving the manuscript quality following my and other reviewers' previous suggestions.

My current suggestions are mostly related to the methodology used in your study.
Namely, the voxel size of 8.96 microns used in micro CT assessment does not provide sufficient details for mouse bone assessment, especially considering the size of individual trabeculae in mice's bones. So, this must be included in the study limitations. For further suggestions, please see: Bouxsein ML et al. J Bone Miner Res. 2010; 25(7):1468-86. doi: 10.1002/jbmr.141.
Moreover, I still consider that it would be great if the graphical representation of the study design could be added to the manuscript since it would be easier for the readers to comprehend all the analysis done in your study, meaning that this could improve the replicability of your study. Considering the rebuttal comment about the technical type of graphic representation, I feel like this should be the author's decision to express themselves in the best way.

Experimental design

Thank you for your diligent efforts in significantly improving the manuscript quality following my and other reviewers' previous suggestions.
I still consider that it would be great if the graphical representation of the study design could be added to the manuscript since it would be easier for the readers to comprehend all the analysis done in your study, meaning that this could improve the replicability of your study. Considering the rebuttal comment about the technical type of graphic representation, I feel like this should be the author's decision to express themselves in the best way.

Validity of the findings

All underlying data have been provided; they are robust and controlled.

Additional comments

Minor English language editing is still required.

---

## Round 0.3 · accepted · Accept

· Academic Editor

Accept

Thank you for addressing all of the reviewer comments.